# Molecular Epidemiology of Enterovirus A71 in Surveillance of Acute Flaccid Paralysis Cases in Senegal, 2013–2020

**DOI:** 10.3390/vaccines10060843

**Published:** 2022-05-25

**Authors:** Ndack Ndiaye, Fatou Diène Thiaw, Amary Fall, Ousmane Kébé, Khadija Leila Diatta, Ndongo Dia, Malick Fall, Amadou Alpha Sall, Martin Faye, Ousmane Faye

**Affiliations:** 1Virology Department, Institut Pasteur de Dakar, 36, Avenue Pasteur, Dakar 220, Senegal; ndack.ndiaye@pasteur.sn (N.N.); fatoulayethiaw@gmail.com (F.D.T.); amary022@hotmail.com (A.F.); ousmane.kebe@pasteur.sn (O.K.); leiladiatta@gmail.com (K.L.D.); ndongo.dia@pasteur.sn (N.D.); amadou.sall@pasteur.sn (A.A.S.); ousmane.faye@pasteur.sn (O.F.); 2Département de Biologie Animale, Faculté des Sciences et Techniques, Université Cheikh, Anta DIOP de Dakar, Dakar 220, Senegal; malickfal@yahoo.fr

**Keywords:** enterovirus A71, environment, AFP, Africa, Senegal

## Abstract

Enterovirus A71 (EV-A71) is a non-polio enterovirus that currently represents a major public health concern worldwide. In Africa, only sporadic cases have been reported. Acute flaccid paralysis and environmental surveillance programs have been widely used as strategies for documenting the circulation of polio and non-polio enteroviruses. To date, little is known about the molecular epidemiology of enterovirus A71 in Africa where resources and diagnostic capacities are limited. To fill this gap in Senegal, a total of 521 non-polio enterovirus isolates collected from both acute flaccid paralysis (AFP) and environmental surveillance (ES) programs between 2013 and 2020 were screened for enterovirus A71 using real-time RT-PCR. Positive isolates were sequenced, and genomic data were analyzed using phylogeny. An overall rate of 1.72% (9/521) of the analyzed isolates tested positive for enterovirus A71. All positive isolates originated from the acute flaccid paralysis cases, and 44.4% (4/9) of them were isolated in 2016. The nine newly characterized sequences obtained in our study included eight complete polyprotein sequences and one partial sequence of the VP1 gene, all belonging to the C genogroup. Seven out of the eight complete polyprotein sequences belonged to the C2 subgenotype, while one of them grouped with previous sequences from the C1 subgenotype. The partial VP1 sequence belonged to the C1 subgenotype. Our data provide not only new insights into the recent molecular epidemiology of enterovirus A71 in Senegal but also point to the crucial need to set up specific surveillance programs targeting non-polio enteroviruses at country or regional levels in Africa for rapid identification emerging or re-emerging enteroviruses and better characterization of public health concerns causing acute flaccid paralysis in children such as enterovirus A71. To estimate the real distribution of EV-A71 in Africa, more sero-epidemiological studies should be promoted, particularly in countries where the virus has already been reported.

## 1. Introduction

The majority of enteroviruses (EVs) are associated with mild infection; however, some of them, such as enterovirus A71 (EV-A71), are neurotropic and represent a major public health concern. First identified from a female patient with encephalitis in California in 1969, EV-A71 is a member of the Enterovirus genus in the Picornaviridae family [1].

EV-A71 is a highly infectious pathogen mainly transmitted through direct person-to-person contact via the feco-oral route or via respiratory secretions [2]. However, children are most often affected during epidemics, and clinical manifestations include, commonly, Hand Foot and Mouth Disease (HFMD) and herpangina and, occasionally, severe and fatal neurological complications including aseptic meningitis, encephalitis and poliomyelitis-like acute flaccid paralysis (AFP) [3,4].

Although there is no US FDA-approved antiviral agent or vaccine against EV-A71 available, three inactivated EV-A71 whole-virus vaccines were approved by the China National Medical Products Administration (NMPA) and are commercially available in China [5]. WHO recommendations suggest that these vaccines based on the EV-A71-C4a subgenotype could be used worldwide as they cross-react with different genotypes of EV-A71 and exhibit a vaccine efficacy of at least 90% against EV-A71-associated HFMD and 100% against EV-A71-associated HFMD with neurologic complications [6,7].

Based on the VP1 protein, which is the most diversified region between EV species, EV-A71 presents a large genetic diversity with seven genogroups (A to G), and the A genogroup represents the prototype strain (BrCr) [8]. Subsequently, various isolates with a surprising genetic link to the BrCr strain were collected in China [9]. With a worldwide distribution, the B and C genogroups are the well-known groups and are classified into subgenotypes B0 to B5 and C1 to C5, respectively [10]. The D and G genogroups are strictly found in India [11], while the E and F genogroups are known to be endemic to Africa and Madagascar [8].

The C genogroup of EV-A71 was first isolated at the end of the 20th century (C1 in 1986 in Australia; C2 in 1995 in Australia; C3 in 2000 in Korea; C4 in 1998 in Taiwan; and C5 in 2005 in Vietnam [12]) and currently presents a worldwide distribution [13]. The C1 subgenotype emerged in Germany in 2015 and was the most prevalent strain in Europe and caused outbreaks with severe neurologic diseases in France, Poland and Spain in children with HFMD [14,15,16]. The C2 subgenotype was also reported in Europe during the past two decades [17] and was associated with severe clinical manifestations, including aseptic meningitis, myelitis, bronchiolitis, herpangina and fatal encephalitis, during the devastating HFMD outbreaks in Taiwan and Australia in 1998 and 1999, respectively [4]. In addition, the C2 subgenotype has been frequently reported in Asia, Europe, North America and South America as responsible for neurologic complications and deaths [4].

In contrast, the C3 subgenotype sporadically emerged in mainland China [18], while the B1 and B2 subgenotypes were present in America and Europe towards the 1970s. However, the B2 subgenotype was introduced to Australia and Japan in 1980 [19]. B1, B2 and B3 appear to be extinct since the representative genotypes have not been found since 1999 in Singapore [1]. The B4, B5 and C4 subgenotypes were previously restricted to Asia [20,21,22]. However, an introduction of the C4 subgenotype has been recently reported in Europe [23]. 

In Africa, a short epidemic reported 70 HIV-infected orphans in Kenya in 2000 and was related to the C genogroup [24]. In addition, several sporadic isolations of the C1 and C2 subgenotypes were reported from AFP cases between 2000 and 2013 in six African countries including Senegal [25], Democratic Republic of the Congo [26], Nigeria [27], Central African Republic [28], Cameroon [29] and Kenya [24]. 

Through the Global Polio Eradication Initiative (GPEI) program, the World Health Organization (WHO) recommends clinical surveillance of polioviruses and non-polio enteroviruses by investigating cases of acute flaccid paralysis (AFP) in children less than 15 years old and has largely established environmental surveillance (ES) from raw sewage samples to complement AFP surveillance [30,31]. To date, little is known about the molecular epidemiology of EV-A71 in African countries where resources, knowledge and diagnostic capacities are limited [25,26,27,28,29,32].

To better understand the epidemiology of EV-A71 in Africa, particularly in Senegal, herein, we provide insights into the temporal distribution of EV-A71 among AFP cases in Senegal over eight years and assess its recent molecular epidemiology.

## 2. Materials and Methods

### 2.1. Data Collection

To investigate the temporal distribution and genetic diversity of EV-A71 in Senegal, a total of 521 non-polio enterovirus (NPEV)-positive isolates detected from 2013 to 2020 through the routine surveillance activities of polioviruses and according to the WHO guidelines [33,34] were retrospectively tested. The isolates were provided by the World Health Organization’s Regional Polio Laboratory in Senegal and included 373 isolates from AFP cases and 148 from ES.

### 2.2. Sample Preparation and NPEV Classification

Primary stools from AFP cases were treated with chloroform to give clarified stool suspensions, while sewage samples were centrifuged and treated with 5M sodium chloride, 29% polyethylene glycol and 22% dextran T40. The mixture was stirred with a magnet and poured into a sterile separation funnel for each sample. Then, fungizone (0.5%), penicillin G (100 UI/mL) and streptomycin (100 mg/mL) were added after chloroform extraction. Viral isolation was performed simultaneously using RD and L20B cell lines. Briefly, a volume of 200 μL of clarified stool suspensions and concentrated sewage specimens were simultaneously inoculated onto both RD and L20B cells lines. Inoculated cells were incubated at 37 °C and monitored daily for cytopathic effect (CPE). If no CPE was observed by 5 days post inoculation (dpi) in both cell lines, a second blind passage was performed. Samples that showed no CPE in both cells after the second passage were classified as negative. Specimens exhibiting complete CPE only on L20B cells or both L20B and RD were classified as suspected for polioviruses and further characterized using intratypic, differential real-time reverse transcriptase polymerase chain reaction (rRT-PCR) assays, while those that produced CPE only on RD cells were classified as NPEV and confirmed using a pan-enterovirus RT-PCR assay [35]. Supernatants from the NPEV-positive isolates were collected, aliquoted and frozen at −20 °C until use.

### 2.3. RNA Extraction and Molecular Detection

RNA was extracted from 200 µL of supernatants of the NPEV-positive isolates using the QIAmp Viral RNA Mini Kit (QIAGEN, Hilden, Germany) according to the manufacturer’s instructions. RNA was eluted in 60 μL of nuclease-free water and stored at −80 °C until testing. NPEV isolates were confirmed by real-time reverse transcriptase polymerase chain reaction (rRT-PCR) using a pan-enterovirus RT-PCR assay [35] with the Light Mix^®^ Modular Enterovirus 500 kit (Roche-Ref 50-0656-96, TibMolBiol, Berlin, Germany) and the qScript™ XLT One-Step RT-PCR (Quanta Bio, Beverly, MA, USA) according to the manufacturer’s instructions. Experiments were performed using the CFX96TM Real-Time PCR system (Bio-Rad, Singapore). In addition, an rRT-PCR assay for specific detection of EV-A71 was used with the same reagents and equipment as previously described [36]. EV-A71-positive RNA was stored at −80 °C until further testing.

### 2.4. Sequencing of the VP1 Gene

The cDNA was synthetized from EV-A71-positive RNA using random hexamer reverse primer with the reversetAid first-strand kit (invitogen). The generated cDNA was then amplified using a nested PCR method targeting the VP1 region followed by Sanger sequencing, as previously described [37]. VP1 sequences were analyzed using the GeneStudio software (GeneStudio ™ Pro, version: 2.2.0.0, 8/11/2011) (San Diego, CA, USA).

### 2.5. Sequencing

EV-A71-positive RNA was also subjected to high-throughput sequencing. The first-strand cDNA was synthesized using the Super-Script III kit (Invitrogen, Carlsbad, CA, USA). In addition, the second-strand DNA was obtained using the Illumina Nextera kit (Illumina, San Diego, CA, USA) and amplified with the Platinum PCR super-mix kit (Invitrogen). The purified cDNA was quantified using the dsDNA High Sensitivity kit on a Qubit 3.0 fluorometer (Thermo Fisher, Waltham, MA, USA). Then, libraries were prepared from the tagmented DNA using the Nextera XT DNA library prep kit (Illumina) according to the manufacturer’s instructions. The purified libraries were also quantified by using the KAPA Library Quant Kit (Kapa Biosystems, Wilmington, MA, USA), following the manufacturer’s instructions. The normalized libraries were pooled, and sequencing was performed with paired-end reads using the MiSeq reagent kit v2 (for 300 cycles) on an Illumina MiSeq instrument (Illumina). The consensus genomes were de novo assembled using the fully open-source EDGE Bioinformatics software [38].

### 2.6. Phylogenetic Analyses

The generated sequences were manually curated using the GeneStudio software (GeneStudio ™ Pro, version: 2.2.0.0) and analyzed using the online Basic Local Alignment Search (BLAST) program (https://blast.ncbi.nlm.nih.gov/Blast.cgi, accessed on 28 February 2022) to compare the sequence homology with previously available sequences. In addition, the genotyping was confirmed using the online RIVM program (http://www.rivm.nl/mpf/enterovirus/typingtool/, accessed on 28 February 2022). Multiple alignments and tree inference were performed using the Muscle and Maximum Likelihood (ML) methods implemented in the MEGA 7.0 program. Two ML phylogenetic trees were inferred based on the 8 complete polyprotein sequences (~7400 bp) and the partial VP1 sequence (~775 bp) of the 9 isolates identified from Senegal, respectively. A total of 22 additional sequences previously available from NCBI were also used. Trees were inferred for 1000 replications, and nodes were supported by the bootstrap values. The evolutionary distances were estimated using the Tamura 3 parameter method. Bootstrap values ≥90 were shown on the tree, which was rooted on midpoints. 

## 3. Results

### 3.1. Temporal Distribution of EV-A71 Isolates

From January 2013 to December 2020, a total of 521 rRT-PCR-confirmed NPEV isolates were identified, including 373 isolates from AFP cases and 148 from sewage samples. Overall, a rate of 1.72% (9/521) tested positive for EV-A71 by rRT-PCR, and all EV-A71-positive isolates were detected in isolates obtained from AFP cases. The annual positivity rates ranged between 1.5% and 10%, with the highest ratio recorded in 2016 (Table 1).

Over this eight-year surveillance period, EV-A71 showed a low prevalence in AFP cases, and the highest detection rate was found in 2016 with 44.4% (n = 4/9). In addition, our data exhibited a seasonal distribution for EV-A71 circulating in Senegal. EV-A71-positive isolates were mainly detected between January and July, and 66% (n = 6) of them were found between February and March, which corresponds to the end of the cold season in Senegal (Figure 1).

### 3.2. Geographical Distribution and Genomic Analysis

The nine EV-A71-positive isolates were distributed in seven out of the 14 regions in Senegal, and 33% (3/9) of them originated from the Diourbel region, which is the second most populated region in Senegal. In addition, all nine sequences included AFP cases and not contacts. All nine EV-A71 sequences were successfully characterized and included one partial sequence of the VP1 gene and eight complete genomes. Two of the complete sequences (14-157 and 15-355) were included in one of our previous studies [39]. BLAST and RIVM analyses showed that the new EV-A71 sequences belonged to the C genogroup, including eight in genogroup C2 and one in genogroup C1. Sequences were submitted to NCBI Genbank (https://www.ncbi.nlm.nih.gov/nucleotide, accessed on 15 March 2022) under the following accession numbers (Table 2).

### 3.3. Phylogenetic Analysis

The ML tree inference confirmed the data from the BLAST and RIVM analyses and showed that seven complete EV-A71 sequences from Senegal belonged to the C2 subgenotype and clustered with isolates from Mauritania in 2014 and Guinea in 2015, while one complete genome sequence grouped with a sequence from the Central African Republic in 2003 that belonged to the C1 subgenotype (Figure 2A). The partial sequence of the VP1 gene belonged to the C2 subgenotype (Figure 2B). These data were supported with bootstrap values ≥ 90 (Figure 2).

## 4. Discussion

AFP and ES surveillance programs are essential strategies widely used by WHO for identification of emerging and re-emerging enteroviruses, such as EV-A71, and better understanding of their molecular epidemiology. However, only few data are currently available on the molecular epidemiology of EV-A71 in Africa, where resources and knowledge are limited. In this study, the temporal distribution of EV-A71 circulating in Senegal and its molecular epidemiology were assessed over eight years through the existing AFP and ES surveillance programs for enteroviruses. 

Although enteroviruses are known to have continuous circulation throughout the year, the seasonal distribution for EV-A71 shown in our study provided a new insight into its epidemiology. Despite the highest detection rate of EV-A71 being recorded during 2016, the circulation of EV-A71 in Senegal was characterized by sporadic cases. Thus, there is a need for more longitudinal studies targeting the circulation of EV-A71, particularly in Africa, to confirm these data. In addition, the highest prevalence of EV-71 during 2016 and in the Diourbel region could be associated with intrinsic environmental factors, such as temperature, as previously reported in Taiwan [40]. Nevertheless, the possible risk factors in this region need to be further investigated. Though no US FDA-approved antiviral agent or vaccine against EV-A71 is currently available [41], sero-epidemiological studies focusing on determination of the disease’s burden in Senegal and evaluation of risk factors associated with its implication in severe infections, such as AFP, could be important at country level.

The absence of EV-A71 among NPEV isolated from sewage samples during this eight-year period could be related to the existence of only two sewage sites currently used for ES in Senegal, with bimonthly sample collections per site. These two ES sites are located in the Dakar region. A multisite and country-wide ES with a regular and weekly collection of sewage samples is important to complement the existing national AFP surveillance and better assess the emergence of EV-A71 in Senegal, particularly in the Diourbel region. ES has already proven its usefulness as an efficient alert system to detect circulation of pathogens in communities, allowing the establishment of rapid and appropriate prevention and control measures to stop circulation [31,42]. 

Our data confirmed the circulation of genogroup C in Senegal, which clustered with isolates from Central and West Africa as previously described [39]. This genetic link between sequences could be associated with possible virus importations from neighboring countries, human immigrations and food circulation between Central and West African countries, as previously reported for other viruses, such as the West Nile virus and coronaviruses [43,44]. The C2 subgenotype was the most prevalent EV-A71 type among AFP cases from Senegal. More attention should be paid to this subgenotype as it has previously been associated with severe neurologic diseases, such as aseptic meningitis, myelitis, bronchiolitis and fatal encephalitis in Asia, Europe, North America and South America [4,16,17]. 

It is also important to promote more experimental studies focusing on the pathogenesis of EV-A71 isolates belonging to the C1 and C2 subgenotypes to better understand their biology and fitness in children who are at high risk.

Most of the currently available EV-A71 sequences are from Europa, Asia and the Americas [45]. Thanks to considerable efforts deployed by the international health organizations for capacity building and sequencing in Africa during the current COVID-19 pandemic [46], African research teams working especially on NPEV could develop sequencing pipelines that allow the generation of more data for a better understanding of the genetic diversity of public health concerns such as EV-A71. Interestingly, the new, complete sequences could also be useful in large phylodynamic studies of EV-71 worldwide and for identification of epitopes or motives that could be targeted for the development of new vaccines or therapeutics or updates of the currently available countermeasures [5,41,47].

## 5. Conclusions

Our study represents the second report on the genetic diversity of EV-A71 in Senegal and is noteworthy for describing the temporal distribution of EV-A71 in children with AFP from Senegal and its recent epidemiology marked by the circulation of the C1 and C2 subgenotypes. A more specific surveillance targeting NPEV and based on country-wide AFP and ES surveillance programs is important for a better understanding of EN-A71’s geographic distribution. In addition, further sero-epidemiological studies are also needed to estimate the real burden of EV-A71 in the country and evaluate the different risk factors associated with its severity in AFP cases. More laboratory experiments could also be promoted to assess the pathogenicity of the identified isolates.

## Figures and Tables

**Figure 1 vaccines-10-00843-f001:**
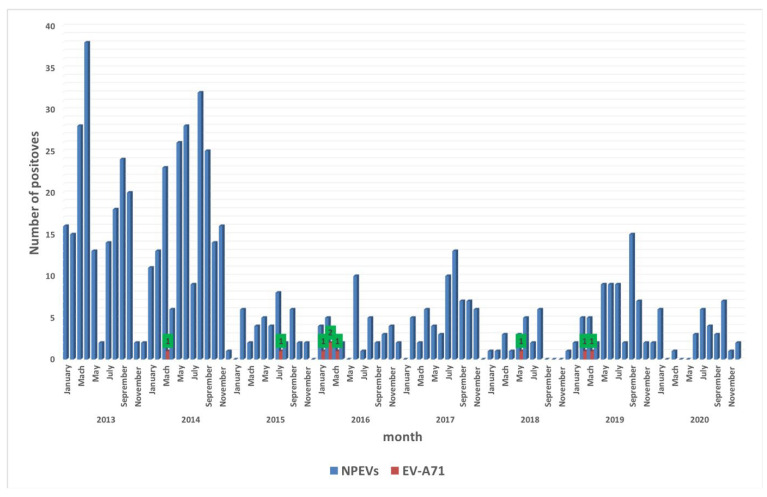
Monthly distribution of NPEV and EV-A71 identified in Senegal from 2013 to 2020. The shaded bars show positive specimens from patients for NPEV (in blue) and EV-A71 (in red). The green boxes with numbers (1 or 2) indicate the number of EV-A71-positive isolates.

**Figure 2 vaccines-10-00843-f002:**
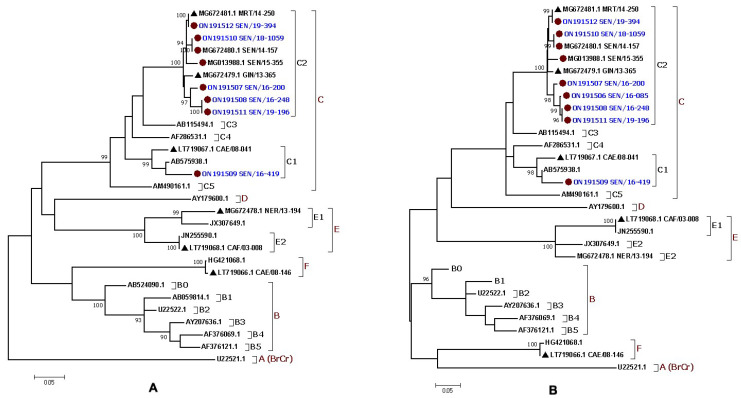
Maximum likelihood (ML) phylogenetic trees based on (**A**) eight complete polyprotein sequences (~7400 bp) and (**B**) nine partial VP1 sequences (~775 bp) from the EV-A71 isolates identified in Senegal. The newly characterized sequences are highlighted in blue. Bootstrap values ≥ 90 are shown on the tree. The small red circle and the small black triangle indicate the strains from Senegal and the other African countries, respectively. The scale bar indicates the distances of the branches. The tree shows that all the Senegalese EV-A71 strains belonged to the genogroup C.

**Table 1 vaccines-10-00843-t001:** Yearly distribution of NPEV and EV-A71 detected in AFP and environmental surveillance programs from 2013 to 2020.

Year of Isolation	Number of NPEV Isolates		Number of EV-A71 Isolates		Ratio of EV-A71-Positive Isolates (%)	
	AFP	ES	AFP	ES	AFP	ES
2013	36	-	-	-	-	-
2014	68	-	1	-	1.5	-
2015	41	24	1	-	2.4	-
2016	40	11	4	-	10	-
2017	63	35	-	-	-	-
2018	23	27	1	-	4.3	-
2019	69	31	2	-	2.9	-
2020	33	20	-	-	-	-
Total	373	148	9	-	2.4	-

**Table 2 vaccines-10-00843-t002:** Description of the newly characterized sequences of EV-A71.

Strain	Years	Locality	Patient’s Gender	Case or Contact	Genogroup (Subgenotype)	Genbank Accession Number	Region	Reference
14-157	2014	Thies	Female	Case	C (c2)	MG672480	Complete polyprotein	[39]
15-355	2015	Kaffrine	Female	Case	C (c2)	MG013988	Complete polyprotein	[39]
16-085	2016	Diourbel	Male	Case	C (c2)	ON191506	Partial VP1 gene	This study
16-200	2016	Dakar	Male	Case	C (c2)	ON191507	Complete polyprotein	This study
16-248	2016	Saint Louis	Male	Case	C (c2)	ON191508	Complete polyprotein	This study
16-419	2016	Kaolack	Male	Case	C (c1)	ON191509	Complete polyprotein	This study
18-1059	2018	Tambacounda	Male	Case	C (c2)	ON191510	Complete polyprotein	This study
19-394	2019	Diourbel	Male	Case	C (c2)	ON191511	Complete polyprotein	This study
19-196	2019	Diourbel	Female	Case	C (c2)	ON191512	Complete polyprotein	This study

## Data Availability

All the data are available in the present manuscript.

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
