# Peer review of "Molecular Epidemiology of Enterovirus A71 in Surveillance of Acute Flaccid Paralysis Cases in Senegal, 2013–2020"

_vaccines, 2022, doi:10.3390/vaccines10060843_

Round 1
Reviewer 1 Report
The authors detected 521 non-polio enterovirus (NPEVs)-positive isolates during AFP case surveillance (373 NPEVs) and environmental surveillance (148 NPEVs) in Senegal from 2013 to 2020. Eight genotype C2 EV-A71 and one genotype C1 EV-A71 were identified, and all from AFP patients. To date, the molecular epidemiology of enterovirus A71 in African regions is poorly understood. Therefore, this manuscript has some practical implications for understanding the molecular epidemiology of EV-A71 worldwide, especially for filling gaps in the African region.
The following comments should be considered for revisions to the manuscript:
1, Because all nine EV-A71 is detected in AFP cases. Therefore, it is recommended to modify the title of the manuscript, Molecular epidemiology of enterovirus A71 in surveillance of acute flaccid paralysis cases in Senegal, 2013-2020.
2, The paper has many grammatical errors, and language errors, such as line 275:”APF cases” should be "AFP cases", which should be carefully revised by the authors.
3, Line 55, Wrong use of references [2-4], it is estimated that it copies other literature and forgets to remove it,
4,Table 2:Note the full spelling of "F" and "M" of Patient's genre
5,Figure 2:Note contains duplicate content, and the description of the method can be placed in MAt&Met, there is no need for duplicates here, and the authors need to indicate which region of the sequence was used to construct the phylogenetic tree and what algorithm is used.
Author Response
Open Review
( ) I would not like to sign my review report
(x) I would like to sign my review report
English language and style
( ) Extensive editing of English language and style required
(x) Moderate English changes required
( ) English language and style are fine/minor spell check required
( ) I don't feel qualified to judge about the English language and style
Yes |
Can be improved |
Must be improved |
Not applicable |
|
Does the introduction provide sufficient background and include all relevant references? |
(x) |
( ) |
( ) |
( ) |
Are all the cited references relevant to the research? |
( ) |
(x) |
( ) |
( ) |
Is the research design appropriate? |
( ) |
(x) |
( ) |
( ) |
Are the methods adequately described? |
( ) |
(x) |
( ) |
( ) |
Are the results clearly presented? |
( ) |
(x) |
( ) |
( ) |
Are the conclusions supported by the results? |
( ) |
(x) |
( ) |
( ) |
Comments and Suggestions for Authors
The authors detected 521 non-polio enterovirus (NPEVs)-positive isolates during AFP case surveillance (373 NPEVs) and environmental surveillance (148 NPEVs) in Senegal from 2013 to 2020. Eight genotype C2 EV-A71 and one genotype C1 EV-A71 were identified, and all from AFP patients. To date, the molecular epidemiology of enterovirus A71 in African regions is poorly understood. Therefore, this manuscript has some practical implications for understanding the molecular epidemiology of EV-A71 worldwide, especially for filling gaps in the African region.
The following comments should be considered for revisions to the manuscript:
1, Because all nine EV-A71 is detected in AFP cases. Therefore, it is recommended to modify the title of the manuscript, Molecular epidemiology of enterovirus A71 in surveillance of acute flaccid paralysis cases in Senegal, 2013-2020.
Response: The title of the manuscript has been modified in the revised version.
2, The paper has many grammatical errors, and language errors, such as line 275:”APF cases” should be "AFP cases", which should be carefully revised by the authors.
Response: These typos have been corrected in the revised manuscript.
3, Line 55, Wrong use of references [2-4], it is estimated that it copies other literature and forgets to remove it,
Response: These references have been corrected in the revised manuscript
4,Table 2:Note the full spelling of "F" and "M" of Patient's genre
Response: This spelling has been edited in the revised manuscript.
5,Figure 2:Note contains duplicate content, and the description of the method can be placed in MAt&Met, there is no need for duplicates here, and the authors need to indicate which region of the sequence was used to construct the phylogenetic tree and what algorithm is used.
Response: The Figure 2 has been corrected in the revised manuscript. Two ML trees were inferred based on the complete polyprotein sequences and the partial VP1 sequences. The materials and methods section and the Table 2 have been revised accordingly.
Reviewer 2 Report
This manuscript by Ndack Ndiaye aimed to report the molecular epidemiology of Enterovirus A71 in Senegal, West Africa, over eight years between 2013 and 2020. A total of 521 non-polio enteroviruses (NPEVs)-positive isolates detected from 2013 to 2020 through the routine surveillance activities of polioviruses were retrospectively tested using real-time RT-PCR. An overall rate of 1.72% (9/521) of the analyzed samples tested positive for Enterovirus A71. Out of the nine new complete sequences obtained in the study, eight belonged to the C2 subgenogroup, while one grouped with previous sequences from the C1 subgenogroup. The C1 subgenogroup has been the most prevalent strain in Europe since 2015, while the C2 subgenogroup has also been reported in Europe during the past two decades. The authors also argued that there is a crucial need to set up specific surveillance programs targeting non-polio enteroviruses at the country and regional levels in Africa for rapid identification of emerging enteroviruses of public health concern and better characterization of the circulation of Enterovirus A71 in children with acute flaccid paralysis. Please refer to the comments below.
Comment 1: The abstract is very poorly written. The research focus is very confusing whether the authors would like to compare the prevalence of Enterovirus A71 in Africa vs. Asian regions or within Africa. The study characterizes the circulation of Enterovirus A71 in Senegal, West Africa, between 2013 to 2020. What is the prevalence of Enterovirus A71 in other regions in Africa?
Comment 2: The introduction is very poorly written. How common is Enterovirus A71 to manifest clinically as herpangina, paralysis, myocarditis, aseptic meningitis, encephalitis, and acute respiratory symptoms such as pulmonary edema and deaths among young children? (lines 43-47). Studies have reported that the Enterovirus A71 commonly causes hand, foot, and mouth disease and, occasionally, associated with severe and sometimes fatal neurologic diseases, including aseptic meningitis, encephalitis, and poliomyelitis-like acute flaccid paralysis (AFP) (Solomon et al., 2010; Chang et al., 2016). The epidemiology, evolution, classification, and disease aspects of Enterovirus A71 are mixed in only one paragraph (lines 38-71). This paragraph should be split into more paragraphs. The Enterovirus A71 is classified into seven genogroups (A-G) and further subdivided into subgenotypes. Which genotype(s) and subgenotype(s) are currently circulating in Africa and why? I noticed that in the discussion, the authors attempted to discuss the subgenotypes which should be in the introduction so the reader could follow. Why is it important to start examining the prevalence of the subgenotypes of Enterovirus A71 in Senegal, West Africa? There is too much crucial information relevant to the study but is not spelled out well, and some are missing.
Comment 3: The material and methods section is kind of confusing. What is the difference between the isolates and the samples? (lines 83-105). Table 1 also summarizes the total number of NPEV isolates with two columns labeled AFP and ENV. What is ENV stands for? I assume that ENV means the environmental samples. The NPEV classification method is a bit confusing. It appears to me that the authors classified the NPEV isolates based on whether the inoculum (stool suspensions or sewage specimens) caused CPE or no CPE on L20B and RD cells. If the specimens exhibit complete CPE on L20B cells or L20B and RD, those specimens are excluded, while those that produced CPE only on RD cells were classified as NPEVs (lines 98-104). Please explain the rationale behind this selection method. CPE is a qualitative tool for preliminary selection. Were there other tests performed to confirm the classification of the NPEVs?
For the subsection VP1 protein sequencing, do you mean VP1 gene sequencing or just VP1 protein sequencing? Do you have the amino acid sequence if it is a VP1 protein sequence? You did perform a reverse transcription of the RNA. Why did you use random primers instead of VP1 gene-specific primers? Enterovirus A71 should be abundant in the supernatant since you propagate the virus in the RD cells.
Comment 4: There is so much information in the discussion section that should be in the introduction section. For example, lines 215-227, 241-254, and 255-259. The information on the listed lines would form a solid foundation to argue the significance of the need for surveillance in other African regions.
Author Response
Open Review
(x) I would not like to sign my review report
( ) I would like to sign my review report
English language and style
( ) Extensive editing of English language and style required
( ) Moderate English changes required
( ) English language and style are fine/minor spell check required
(x) I don't feel qualified to judge about the English language and style
Yes |
Can be improved |
Must be improved |
Not applicable |
|
Does the introduction provide sufficient background and include all relevant references? |
( ) |
( ) |
(x) |
( ) |
Are all the cited references relevant to the research? |
( ) |
( ) |
(x) |
( ) |
Is the research design appropriate? |
( ) |
(x) |
( ) |
( ) |
Are the methods adequately described? |
( ) |
( ) |
(x) |
( ) |
Are the results clearly presented? |
( ) |
(x) |
( ) |
( ) |
Are the conclusions supported by the results? |
( ) |
( ) |
(x) |
( ) |
Comments and Suggestions for Authors
This manuscript by Ndack Ndiaye aimed to report the molecular epidemiology of Enterovirus A71 in Senegal, West Africa, over eight years between 2013 and 2020. A total of 521 non-polio enteroviruses (NPEVs)-positive isolates detected from 2013 to 2020 through the routine surveillance activities of polioviruses were retrospectively tested using real-time RT-PCR. An overall rate of 1.72% (9/521) of the analyzed samples tested positive for Enterovirus A71. Out of the nine new complete sequences obtained in the study, eight belonged to the C2 subgenogroup, while one grouped with previous sequences from the C1 subgenogroup. The C1 subgenogroup has been the most prevalent strain in Europe since 2015, while the C2 subgenogroup has also been reported in Europe during the past two decades. The authors also argued that there is a crucial need to set up specific surveillance programs targeting non-polio enteroviruses at the country and regional levels in Africa for rapid identification of emerging enteroviruses of public health concern and better characterization of the circulation of Enterovirus A71 in children with acute flaccid paralysis. Please refer to the comments below.
Comment 1: The abstract is very poorly written. The research focus is very confusing whether the authors would like to compare the prevalence of Enterovirus A71 in Africa vs. Asian regions or within Africa. The study characterizes the circulation of Enterovirus A71 in Senegal, West Africa, between 2013 to 2020. What is the prevalence of Enterovirus A71 in other regions in Africa?
Response: The abstract has been revised for more clarity and better understanding of the objectives of our study as suggested by the reviewer.
Comment 2: The introduction is very poorly written. How common is Enterovirus A71 to manifest clinically as herpangina, paralysis, myocarditis, aseptic meningitis, encephalitis, and acute respiratory symptoms such as pulmonary edema and deaths among young children? (lines 43-47). Studies have reported that the Enterovirus A71 commonly causes hand, foot, and mouth disease and, occasionally, associated with severe and sometimes fatal neurologic diseases, including aseptic meningitis, encephalitis, and poliomyelitis-like acute flaccid paralysis (AFP) (Solomon et al., 2010; Chang et al., 2016). The epidemiology, evolution, classification, and disease aspects of Enterovirus A71 are mixed in only one paragraph (lines 38-71). This paragraph should be split into more paragraphs. The Enterovirus A71 is classified into seven genogroups (A-G) and further subdivided into subgenotypes. Which genotype(s) and subgenotype(s) are currently circulating in Africa and why? I noticed that in the discussion, the authors attempted to discuss the subgenotypes which should be in the introduction so the reader could follow. Why is it important to start examining the prevalence of the subgenotypes of Enterovirus A71 in Senegal, West Africa? There is too much crucial information relevant to the study but is not spelled out well, and some are missing.
Response: The introduction section has been revised according to the reviewer’s comments.
Comment 3: The material and methods section is kind of confusing. What is the difference between the isolates and the samples? (lines 83-105). Table 1 also summarizes the total number of NPEV isolates with two columns labeled AFP and ENV. What is ENV stands for? I assume that ENV means the environmental samples. The NPEV classification method is a bit confusing. It appears to me that the authors classified the NPEV isolates based on whether the inoculum (stool suspensions or sewage specimens) caused CPE or no CPE on L20B and RD cells. If the specimens exhibit complete CPE on L20B cells or L20B and RD, those specimens are excluded, while those that produced CPE only on RD cells were classified as NPEVs (lines 98-104). Please explain the rationale behind this selection method. CPE is a qualitative tool for preliminary selection. Were there other tests performed to confirm the classification of the NPEVs? For the subsection VP1 protein sequencing, do you mean VP1 gene sequencing or just VP1 protein sequencing? Do you have the amino acid sequence if it is a VP1 protein sequence? You did perform a reverse transcription of the RNA. Why did you use random primers instead of VP1 gene-specific primers? Enterovirus A71 should be abundant in the supernatant since you propagate the virus in the RD cells.
Responses: The expression “isolate” was standardized in the manuscript to describe NPEV-positive supernatants while “sample” indicate the primary specimens.
The expression “ENV” was corrected corrected in the revised manuscript as “ES” meaning the environmental surveillance.
The NPEV classification method was edited in the revised manuscript for more clarity. The NPEV isolates were confirmed by real-time RT PCR.
The title of the section 2.4 has been modified in the revised manuscript. We used random primers herein because VP1 gene-specific primers did not give a good result in contrary random primers with the reversetAid first-strand kit in our experiments
Comment 4: There is so much information in the discussion section that should be in the introduction section. For example, lines 215-227, 241-254, and 255-259. The information on the listed lines would form a solid foundation to argue the significance of the need for surveillance in other African regions.
Response: The Discussion section has been edited in the revised manuscript and some of the general information have been m
Round 2
Reviewer 2 Report
I am satisfied with the revised manuscript except for the abstract. Please modify lines 14-19 and combine them into one or two sentences. The second and the fourth sentences do not necessarily have to be in the abstract.
Author Response
Responses to reviewer’s comments
Reviewer 2:
Comments and Suggestions for Authors
I am satisfied with the revised manuscript except for the abstract. Please modify lines 14-19 and combine them into one or two sentences. The second and the fourth sentences do not necessarily have to be in the abstract.
Response: The abstract has been edited in the revised manuscript and the second and the fourth sentences have been removed from the abstract.